# The rhizosphere of *Phaseolus vulgaris* L. cultivars hosts a similar bacterial community in local agricultural soils

**Griselda López Romo** [1], **Rosa Isela Santamaría**[1], **Patricia Bustos** [1], **Francisco Echavarría** [2], **Luis Roberto Reveles Torres**[2], **Jannick Van Cauwenberghe**[3], **Víctor González**[1]*

**1** Centro de Ciencias Genómicas, Universidad Nacional Autónoma de México, Cuernavaca, Morelos, Mexico, **2** Instituto Nacional de Investigaciones Forestales, Agrícolas y Pecuarias, Calera, Zacatecas, Mexico, **3** Institute of Biodiversity, Faculty of Biological Sciences, Cluster of Excellence Balance of the Microverse, Friedrich Schiller University Jena, Jena, Germany

* vgonzal@ccg.unam.mx

## Abstract

This study aimed to investigate the impact of various common beans (*Phaseolus vulgaris* L.) cultivars on the bacterial communities in the rhizosphere under local agricultural conditions. Even though the differences in cultivation history and physicochemical properties of nearby agriculture plots, the bacterial community in the bulk soil was quite similar and more diverse than that of the rhizosphere. The bacterial community of the rhizosphere was closely similar between Black and Bayo common bean cultivars but differs from Pinto Saltillo common beans collected in a different season. A shared bacterial group within the rhizosphere community across cultivars and specific taxa responding uniquely to each cultivar suggests a balance between responses to soil and plant cultivars. Nevertheless, rhizosphere composition was substantially influenced by the pre-existing soil bacterial community, whose diversity remained consistently similar under the studied field conditions. These findings provide a more comprehensive characterization of the rhizosphere across a limited range of domesticated common beans and agronomic soils that can be expanded to more common bean cultivars and soils to guide appropriate field interventions.

## Introduction

The rhizosphere, the soil region surrounding plant roots, has been extensively studied [1]. Microbes associated with the root zone profoundly affect the growth, health, and productivity of the cultivated plants. Therefore, understanding the microbiome diversity in local agricultural settings is essential for sustainable agriculture. Earlier research on model plants, such as *Arabidopsis* and *Lotus*, as well as in agriculturally important plants, such as rice, maize, wheat, soybean, and tomato, has established that the rhizosphere microbiome assembles from the soil under the influence of plant root exudates and rhizodeposition [2–6]. Concurrently, it has been demonstrated that the richness and diversity of microorganisms are typically high in bulk soil but low in cultivated plants [7–9]. Thus, the interplay between the plant and bacterial

**Data availability statement:** All relevant data are within the manuscript and its Supporting Information files and in the GitHub repository: https://github.com/vgonzal54/common-bean-rhizosphere.

**Funding:** PAPIIT-UNAM IN215908 for VG supported this study. GLR is a doctoral student from Programa de Doctorado en Ciencias Biomédicas, Universidad Nacional Autónoma de México (UNAM), and received a fellowship (No. CVU 746216) obtained from CONAHCYT. J.V.C. was also supported by the Deutsche Forschungsgemeinschaft (DFG, German Research Foundation) under Germany's Excellence Strategy–EXC 2051–Project-ID 390713860 and by the European Research Council (ERC) Consolidator grant 865694: DiversiPHI, the Deutsche Forschungsgemeinschaft (DFG, German Research Foundation) under Germany's Excellence Strategy–EXC 2051–Project-ID 390713860, and the Alexander von Humboldt Foundation in the context of an Alexander von Humboldt-Professorship founded by German Federal Ministry of Education and Research. This research was supported by the PAPIIT-UNAM IN215908 for VG. The funders had no role in study design, data collection and analysis, decision to publish, or preparation of the manuscript.

**Competing interests:** NO authors have competing interests Enter: The authors have declared that no competing interests exist.

communities in bulk soil is a crucial factor in determining the composition of the rhizosphere microbiome and other compartments, such as the closest microbial community adhered to the plant root at the rhizoplane and the endophyte community in the endosphere [3,10]. Despite this, the variety of plants in wild and domestic locations makes it challenging to find consistent patterns in the composition, structure, and function of the rhizosphere microbiome [11,12]. Moreover, the spatial distribution of soil and rhizosphere microbiomes in cultivated plants on large and local scales has scarcely been addressed [13].

Nitrogen fixation symbiosis between *Rhizobium* and diverse leguminous species of the Fabaceae family is an outstanding example of the close relationship between plant roots and bacteria [14]. This symbiosis significantly contributes to nitrogen incorporation into the biosphere through the formation of N-fixing nodules in the roots. Domestication of several Fabaceae species for agriculture and human consumption dates back to at least 10,000 years and continues with modern breeding techniques [8,15,16]. Selected genetic features of cultivated plants, such as flowering, root architecture, pathogen resistance, and drought resistance, likely play a role in favoring plant adaptation to variable agricultural conditions [17]. There are a substantial number of cultivated common bean accessions in major common bean-producing countries around the globe, which demonstrates the extensive genetic diversity of this crop and its capacity to adapt to regional agricultural conditions [18]. However, knowledge of the microbial species residing in the rhizosphere and their role in common bean adaptation is poor, and few cultivars have been studied in this regard [19,20].

There is experimental evidence of a reduction in the diversity of the bacterial community from wild to cultivated plants, likely because of the rhizosphere effect and domestication [21–23]. Cultivation history and agricultural practices influence the composition, diversity, and functional features of the rhizosphere microbiome. Crop rotation favors microbial diversity, in contrast to long-term monoculture and tillage practices that reduce the diversity of the soil bacterial population [24–26]. In the short term, ammonia fertilization favors the productivity of common beans. However, in the long term, fertilizer can deplete the diversity of the rhizobial community and the microbiome community structure [27,28]. Furthermore, fungal pathogens such as *Fusarium oxysporum* and *Rhizoctonia solani*, change the structure, composition, and gene expression profiles of the rhizosphere and endophytic bacterial community of common bean [29,30]. In disease-suppressive soils, some bacterial genera, such as *Flavobacterium*, *Pseudomonas*, and *Bacillus,* have been shown to antagonize pathogenic fungi [29–31]. As a result, the interaction of physical, chemical, and biological factors in agricultural soil influences the profile of microbiome communities under the genotype of cultivated plants.

The bacterial rhizosphere community of cultivated common beans indicates species dominance from Pseudomonadota, Actinomycetota, and Bacteroidota [22]. Additionally, the response of the bacterial rhizosphere community across various common bean varieties and agricultural soils suggests high variability in the composition of the bacterial community and a reduced common and persistent core set of genera [13]. Remarkably, contrasting soil types influenced the rhizosphere community, as revealed by the differential composition and abundance detected at lower taxonomic ranks, such as families and genera [22]. However, the effects of unrelated common bean cultivars growing under local agricultural conditions have scarcely been addressed [13]. Therefore, we believe that examining different common bean cultivars in the same agricultural locality may enable us to accurately compare the responses of bulk soil bacterial communities to the presence of plants and to highlight their differences.

Our research primarily focused on the Pinto Saltillo (*P. vulgaris L.*) cultivar, a widely cultivated and consumed bean variety in northern Mexico [32,33]. We then evaluated and compared the bacterial communities in the soil and rhizosphere of Pinto Saltillo with those of nine Black and Bayo bean cultivars grown in the same agricultural region using high-resolution

metagenomic methods. Our comparative study revealed a common bacterial group in the rhizosphere of the cultivars but also specific taxa depending on the cultivar, influenced by the pre-existing soil bacterial community. Moreover, our results showed a consistent group of bacterial genera belonging to Pseudomonadota that were significantly enriched in the rhizospheres of the common bean cultivars grown in local fields.

## Materials and methods

### Experimental sampling

This study was performed at the Instituto Nacional de Investigaciones Forestales, Agrícolas, and Pecuarias (INIFAP) experimental field in Zacatecas, Mexico. Two adjacent agricultural plots, one referred to as "N" for "non-agricultural," which had no history of cultivation but had been used for tillage tests with agricultural machinery for several years [22°54'20.932 "N 102°39'29.504 "W]. The second plot, denoted as "A" or "agricultural," had a history of cultivation with a crop rotation system (corn, beans, and chili) for an extended period [22°54'23.8 "N 102°39'30.4 "W] (S1 Fig). The total area of the plots was 0.34 and 0.42 hectares for sites N and A, respectively. Both sites were planted with the common bean cultivar "Pinto Saltillo" in July 2020.

In each plot ("N" and "A"), we established a grid of 6 × 12 meters, which was further divided into 18 quadrants of 2 × 2 meters (S1 Fig). Bulk soil samples were collected from ten these quadrants for physicochemical analyses. Bulk soil samples (before planting) and rhizosphere samples post-planting were obtained from ten quadrant intersections labeled with letters A to J in both plots, ensuring that the same points in bulk soils corresponded to the locations where bean seeds were sown. Rhizosphere soil was collected after 40-45 days post-planting at the flowering stage. Bulk soil (stage 1) and rhizosphere (stage 2) samples were collected from nine plants for the metagenomic experiments. All samples were stored at -20°C until DNA extraction, but only 17 samples were processed for DNA purification and subsequent metagenomic experiments (S1 Table). Bulk soil samples were collected using metal core augers (10 cm length and 4.3 cm diameter) at a depth of 10 cm. Twenty samples were collected per plot, ten of which were used to determine the physicochemical characteristics of the soil.

This study defined the rhizosphere as soil that was firmly attached to the root [3]. Therefore, we removed the soil surrounding the roots, leaving only the tightly adhered soil. The roots were shaken vigorously in a bag, transferred to a conical tube with SM buffer (100 mM NaCl, 8 mM MgSO4· 7H2O, and 50 mM Tris-Cl, pH 7.5), and shaken vigorously again using a vortex. Subsequently, the roots were removed, the tubes were centrifuged for 10 min at 10,000 rpm, and the supernatant was discarded. Soil pellets were used for DNA purification.

In September 2021, nine additional bean cultivars were selected from the agricultural fields of Zacatecas, Mexico, to compare their bacterial rhizosphere communities with those of Pinto Saltillo bean (S1 Table). The nine cultivars were part of an experiment on the morphology and phenology of 50 bean cultivars conducted by INIFAP (S2 Table). These crops were planted 1.9 km from fields A and N, where Pinto Saltillo was planted, in 2020 [22°54'51.2″ N, 102°39'56.0″ W]. The selected cultivars included five black bean cultivars (Black Creole V10, V2, V4, Black San Luis V26, and Black bean V45) and four Bayo bean cultivars (Peruvian V16, Bayo V22, V47, and Flor de mayo V43). Bulk soil and rhizosphere samples were collected as described above for the Pinto Saltillo cultivar, and the phenology of the selected cultivars was registered (S2 Table)

### Physicochemical characteristics of the soil

The physicochemical properties of the soil were analyzed at the INIFAP Zacatecas Soil Laboratory following the protocols outlined in the Mexican Official Standard (NOM-021-RECNAT-2000),

which provides specifications for soil fertility, health, and classification [34]. Based on these analyses, soil samples from plots A and N were classified as silty clay loam, clay loam, silty clay, and clay, consistent with the commonly described classification for this soil type [22,35].

Chemical characteristics related to soil fertility of the eleven meta-samples were classified as medium alkaline (pH 8-8.49), with a medium proportion of organic matter (1.39 - 3.65%), low inorganic nitrogen (4.67 - 20.97 mg/kg), and with a high level of phosphorus determined with the Olsen method in the case of the agricultural plots (15.98 - 20.47 mg/kg) and a low to medium level in the non-agricultural plots (15.98 - 24.47 mg/kg). The chemical and physical properties are described in S3 Table, and the differences between the A and N sites, as determined by PCA, are presented in S2 Fig PERMANOVA, using Euclidian distances with the Vegan package (v2.6.8), indicated statistically significant differences between A (Agricultural) and N (Non-agricultural) samples regarding the physicochemical properties of the soils ($R^2 = 0.64$; F = 16.11; P = 0.002).

## DNA purification

The bulk soil or rhizosphere (250 mg) was used for DNA extraction using the DNeasy PowerSoil DNA Isolation Kit (Qiagen), following the manufacturer's protocol, and stored at -20°C until analysis. For the Pinto Saltillo experiment, 17 samples were sequenced: three bulk soil samples from plot A, three bulk soil samples from plot N, six rhizosphere samples from plot A, and five rhizosphere samples from plot N (S1 Table). Three DNA samples from the rhizosphere of each Black and Bayo bean cultivars were processed along with three DNA samples corresponding to the bulk soil. All samples were sent to Macrogen (Korea) for shotgun sequencing using the TruSeq kit on an Illumina NovaSeq, with a sequence length of 150 bp pair-end reads.

## Metagenomic sequencing and taxonomic assignment

An average of 94,094,934 and 81,250,279 sequence reads was obtained for the bulk soil and rhizosphere samples, respectively (S4 Table). Sequences were evaluated using FastQC v0.11.8 [36], and quality scores (Q) greater than 20 were processed using TrimGalore v0.6.4R [37]. High-quality reads were taxonomically classified using the Kraken2 program with default parameters: confidence 0.0, minimum-hit-groups 2 [38], and the PlusPFP database (Standard plus protozoa, fungi, and plant, fixed from 12/2/2020 version). In bulk soils, approximately 20% was assigned to known taxa by Kraken2. Conversely, for rhizosphere samples, a median of 77% of the total readings was classified, most of which belonged to the domains Bacteria (97% and 99% for the bulk soil and rhizosphere, respectively), Archaea (0.42, 0.01%), Eukaryota (1.2, 0.2%), and viruses (0.02, 0.01%) (S3 Fig). As anticipated, Bacteria was the most abundant domain and the focus of this study (S3 Fig and S4 Table). Kraken2 parameters and database choice were selected in agreement with the aim to capture a broader range of genomic sequences for the identification of a greater variety of taxa (S1 Supporting Methods; S4 to S8 Figs). Kraken2 taxonomic classification at the species level was used for rarefaction modeling and diversity analysis. Taxonomic classification of the Kraken2 outputs were actualized to the International Code of Nomenclature of Prokaryotes (INCP) adopted by the NCBI Taxonomy [38,39].

Rarefaction curves without replacement were generated using the Vegan package (v2.6.8) in R (v4.3.3) with a recursive function [40] (S1 Supporting Methods) (S9 Fig).

The Phyloseq package (v1.46.0) was used to estimate relative abundance using the subset taxa function. The same package was also employed to estimate alpha diversity (utilizing the estimate_richness functions) based on the number of observed taxa, Chao1, and Shannon and

Simpson indices. Beta diversity was assessed using the Bray-Curtis distance method (utilizing distance and ordinate functions) [41]. Differential abundance estimates were calculated using the DESeq2 package (v1.42.1) [42] and visualized using the EnhancedVolcano (v.1.20.0) package in R [43]. To address the compositional structure of the metagenomes, the relative abundance matrix of taxa classified by Kraken2 was re-calculated with the Centered Log-Ratio Transformation (CLR) using the Microbiome (v1.22.0) and Compositions (v2.0.8) packages. The dissimilarity between samples was calculated by the Aitchinson distance using the CLR data, and PCoA and statistical analyses were performed as described in the next section and in S1 Supporting Methods and S10–S16 Figs.

## Statistical tests

Permutational multivariate analysis of variance (PERMANOVA) was performed with 999 permutations using the distance and adonis2 functions from the Vegan package (v2.6.8) [40]. Wilcoxon and Student's t-tests were performed using Stats in R (v4.3.3) [44]. PCA and PCoA analyses were performed using the ordinate function of the Phyloseq package (v1.46.0) [41]. Dispersion analysis was done with the betadisper and permutest functions from the Vegan package (v2.6.8). Venn diagrams and upset R plots were done with ggven package (v0.1.10) and UpSetR package (v1.4.0).

## Results

### The diversity of the bulk soil and rhizosphere bacterial communities was distinct

Our first experiment aimed to discern the effect of agriculture versus a non-agricultural plot on the bacterial diversity associated with the bulk soil and rhizosphere of common bean (*P. vulgaris* cv. Pinto Saltillo). The experiment was performed in two plots: one with previous agricultural use (site A), and the other with no prior agricultural activity (site N). At both sites, bulk and rhizosphere soil samples were obtained and processed to obtain metagenomic sequences with high coverage and depth (Material and Methods; Supporting Methods; S3 Fig; S4 Table).

The bulk soil samples collected from plots A and N were significantly different in terms of their physical and chemical compositions (PERMANOVA $R^2 = 0.64161$, $F = 16.112$, $P = 0.002$; S3 Table; S2 Fig). However, they showed similar bacterial diversity in both the soil and the rhizosphere (Fig 1). There were no substantial differences in the Chao, Shannon, and inverse Simpson diversity indices between the bulk soils of sites A and N (Fig 1A). Thus, in these comparisons, the cultivation history has little relevance in the bacterial diversity in the agriculture plots used in the study.

Although the Chao1 values were similar between bulk soil and the rhizosphere bacterial communities of sites A and N, the diversity measured by the Shannon index was notably lower (H' = 4.05 for site A and 4.07 for site N) in the rhizosphere than in the bulk soil at the two sites (Fig 1A). Therefore, the results suggest that the diversity of the bacterial community in the bulk soil before planting surpassed that in the rhizosphere (Wilcoxon P = 0.024 and 0.036 for A and N within-site comparisons). Furthermore, the inverse Simpson index of the bacterial community in the bulk soil was greater than that of the rhizosphere samples (Fig 1A). Thus, the bacterial community was more homogeneous in the soil than in the rhizosphere, and dominance of some taxa could be expected in the rhizosphere.

To assess the differences in diversity between the bulk soil and rhizosphere communities, we calculated the beta diversity based on Bray-Curtis dissimilarity. Principal coordinate analysis (PCoA) showed that neither bulk soil nor the bacterial rhizosphere communities between sites

**A**

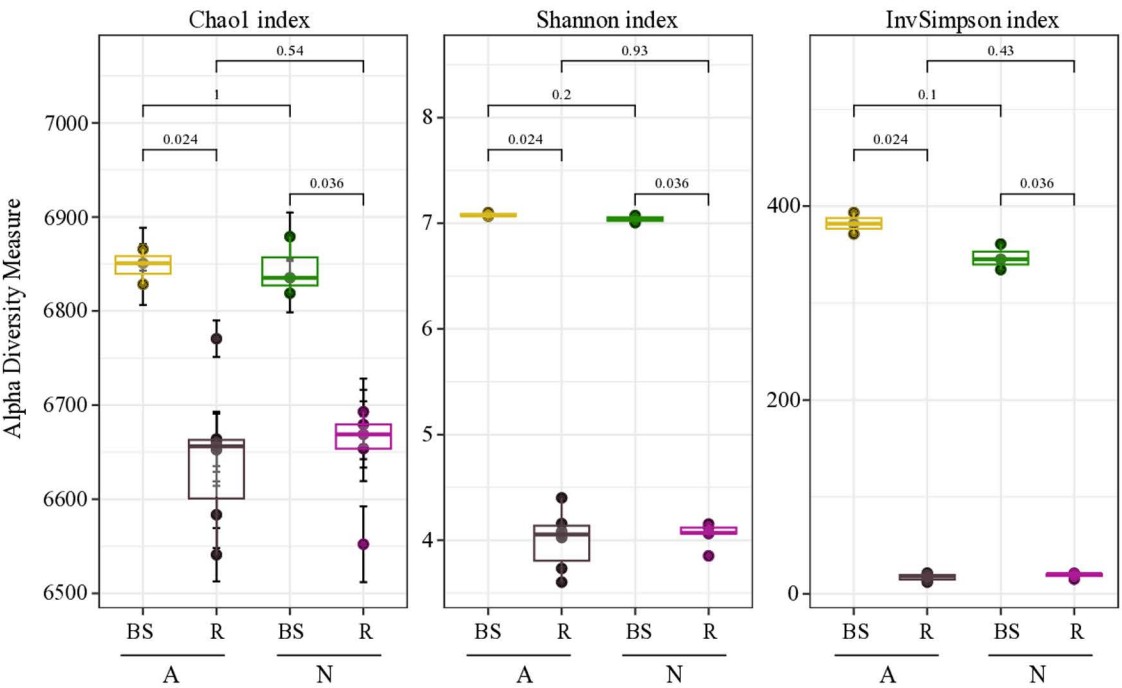

**B**

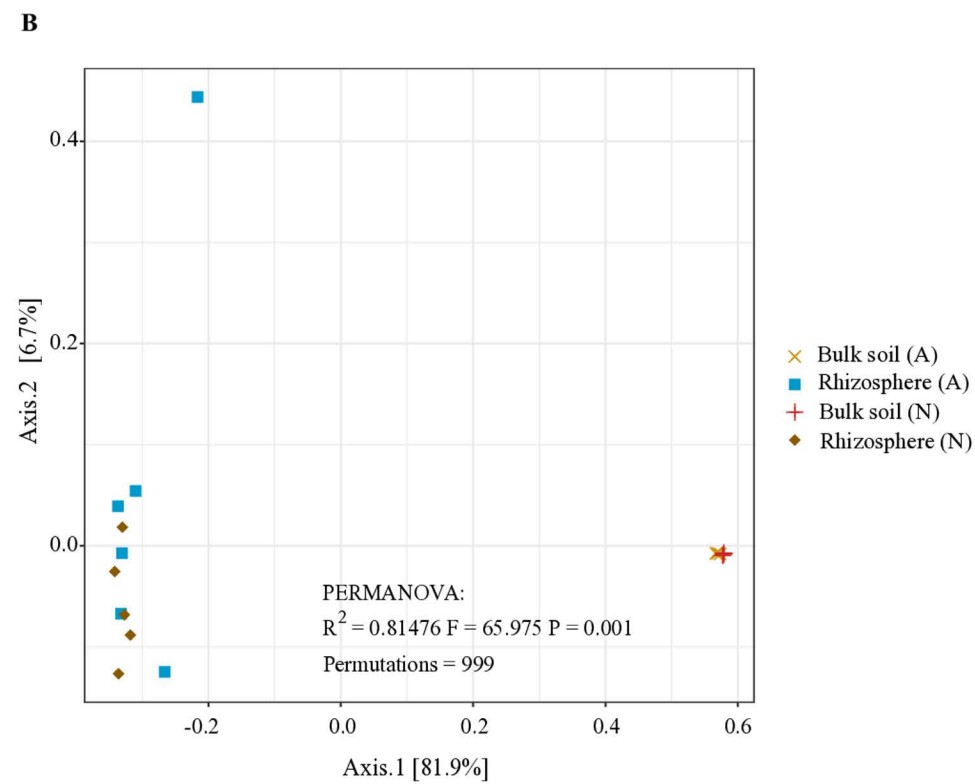

**Fig 1. Bacterial diversity in bulk soil and rhizosphere.**

A and N were significantly different, except for the sample labeled AH, which was substantially different from the rest of the samples. In contrast, there were differences in beta diversity between the bulk soil and rhizosphere communities, which was explained by 81.9% of the variance in the PCoA (Fig 1B, S5 Table, $R^2 = 0.81$, F = 68.9, PERMANOVA, $P < 0.001$). Therefore, the beta diversity patterns revealed that the soil and rhizosphere bacterial communities were distinct and did not overlap. Furthermore, beta diversity analysis confirmed that cultivation history had a marginal effect on the bacterial diversity of the bulk and rhizosphere communities from the A and N sites.

## Pseudomonadota was the dominant phylum in the rhizosphere community

To assess the composition of the soil and rhizosphere bacterial communities, taxon ranks were assigned to metagenomic reads using Kraken2 [45]. Bulk soil samples from sites A and N had similar taxa composition. In the bulk soil, the phyla Actinomycetota and Pseudomonadota accounted for the highest percentages of 54.37% and 58.78% of the classified readings and 40.41% and 35.31% for the A and N sites, respectively (S17 Fig). Members of the phyla Bacillota, Planctomycetota, and Bacteroidota were less abundant (2.89% and 3.10% at Sites A and N, respectively). Among the most abundant genera in the soils were *Streptomyces* (9.91%), *Nocardiodes* (4.95%), *Micronospora* (2.78%) of phylum Actinomycetota; and *Sphingomonas* (2.74%), *Bradyrhizobium* (2.48%), and *Pseudomonas* (1.77%), of the Pseudomonadota (Fig 2A). A minor proportion of the reads was assigned to Actinomycetota (1.02%) and Bacteroidota (0.57%) (S3 Fig). When the proportions of the taxonomic categories within Pseudomonadota were evaluated, the class Gammaproteobacteria (79.24%) and genus *Pseudomonas* dominated the read assignments, and in a lesser proportion, other Alphaproteobacteria and Betaproteobacteria (13.78% and 5.11%, respectively) (S18 Fig). The genus *Pseudomonas* dominated the rhizosphere community and accounted for the highest proportion (60.69%). A set

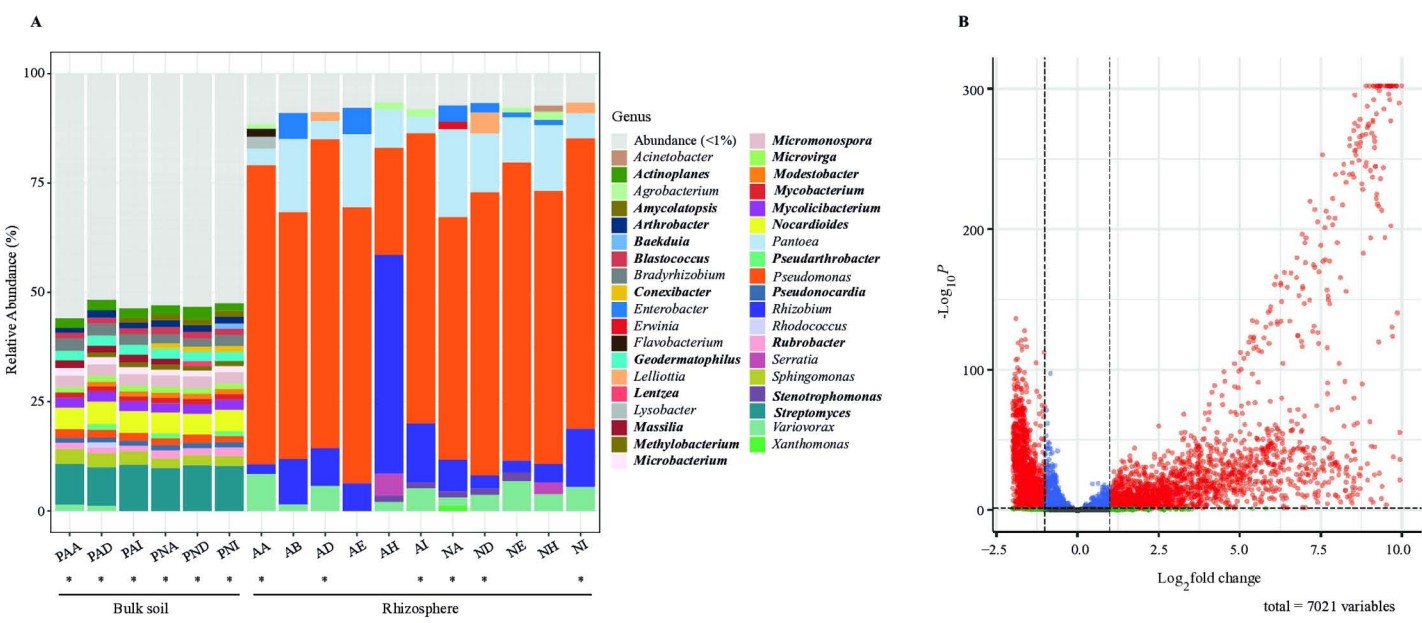

**Fig 2. Relative abundance and bacterial taxonomic composition of the bulk soil and Pinto Saltillo cultivar rhizosphere.** The righThe right inset shows the bacterial genera with abundance > 1%. Genus names in bold correspond exclusively to bulk soil. (B) Volcano plot of differential abundance expressed as log2 (fold change, x-axis) and its statistical significance p-value (-log10, y-axis). A total of 7,021 species determined by Kraken2 were tested for differential abundance, but only 4,856 showed significant changes (p < 0.05), as indicated by red dots. Parallel dashed lines delimited the area containing less significant changes (p > 0.05, log2-fold < 1). This analysis was evaluated using DESeq2. inset.

of six other genera that represented > 1% of the sequence reads, including *Rhizobium* (11%, Alphaproteobacteria), *Pantoea* (10%, Gammaproteobacteria), *Variovorax* (4%, Betaproteobacteria), *Enterobacter* (2%, Gammaproteobacteria), *Serratia* (1%, Gammaproteobacteria), and *Lelliotia* (1%, Gammaproteobacteria), accounted for 30% of the assigned reads (Fig 2A). The remaining phyla and genera were present in low proportions (< 1%) and were composed of Alphaproteobacteria and unknown bacterial taxa. Differences in the bacterial compositions of the soil and rhizosphere communities were observed across all taxonomic hierarchies.

## Shifts in bacterial taxa abundance from the bulk soil to the rhizosphere

To assess the disparities between the bacterial communities in the bulk soil and the rhizosphere, we conducted a differential abundance analysis with DESeq2, combining the bulk soil and rhizosphere taxa from the N and A sites (Fig 3). A total of 4,910 (69%) taxa at the species level according to Kraken2, exhibited changes in their relative abundance upon the presence of the common bean plant (Fig 2B; S6 Table). By performing separate DESeq2 analyses for agricultural (A) and non-agricultural (N) sites, we identified approximately 73% (3,697) common species between N and A sites, 14% (686) of N samples, and 12% (597) belonging to A samples (S16 Fig). These results suggest that the bulk soil might have a common bacterial community that responds to the presence of plants but maintains a group of specific taxa at each site.

Next, we focused on identifying the taxa that substantially increased their representation in the rhizosphere community compared with the bulk soil by more than two Log2fold. Nearly half of the combined set of taxa of N and A sites (53%, 2604/4910) presented this increase. Remarkably, most of these taxa (1922, 39.6%) belonged to Pseudomonadota, with smaller proportions assigned to Bacillota (209, 4.30%) and Bacteroidota (395, 8.13%). Pseudomonadota reads were taxonomically assigned to 321 genera, whereas Bacillota and Bacteroidota reads were assigned to 70 and 113 genera, respectively.

Notably, sequence reads taxonomically assigned to the genera *Pantoea*, *Lelliottia*, *Enterobacter*, *Erwinia*, *Pseudomonas*, *Serratia*, *Acinetobacter*, and *Scandinavium*, all belonging to the class Gammaproteobacteria, showed 7-to 14-fold increase in abundance. Among the Alphaproteobacteria, genera with significant Log2 fold changes included *Rhizobium*, *Agrobacterium*, *Neorhizobium*, *Sinorhizobium*, and *Ensifer*, all enriched 3-to 12-fold. Additionally, the genera *Variovorax* and *Achromobacter*, which belong to Gammaproteobacteria, displayed seven- and four-fold increases in abundance, respectively. Only a few species affiliated within a single bacterial genus significantly increased in proportion to the rhizosphere (S19 Fig). In *Rhizobium*, only three of the 46 taxa at the species level assigned by Kraken2 accounted for the highest sequence read abundance in the rhizosphere. This was also evident from the increased log2-fold change from three to ten for every *Rhizobium* species, according to Kraken2 (S19A Fig). The same pattern was observed in other rhizosphere bacteria, such as *Pseudomonas* and *Variovorax* (S19B and C Fig). In contrast, 1,169 taxa (24.07%) belonging to Actinobacteria species in the bulk soil exhibited significant decreases in the rhizosphere community. Other taxa from Pseudomonadota (10.36%), Bacillota (3.67%), Cyanobacteriota (2.24%), and Bacteroidota (1.17%), represented by 503, 178, 109, and 57 taxa, respectively, also showed a reduction in their presence in the rhizosphere (Fig 2B and S6 Table). Altogether, these results indicate that the common bean cultivar Pinto Saltillo plants significantly influence the rhizosphere community, perhaps by stimulating the growth of a specific array of bacterial species.

## Common bean cultivars exhibited similar bacterial communities

To determine the differences in the bacterial community of the rhizosphere of the Pinto Saltillo cultivar compared to other common bean cultivars, we selected nine common bean

**A**

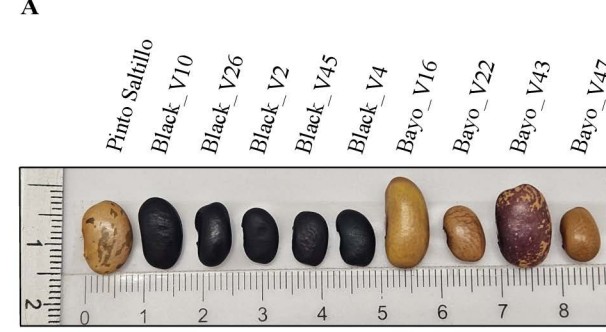

**B**

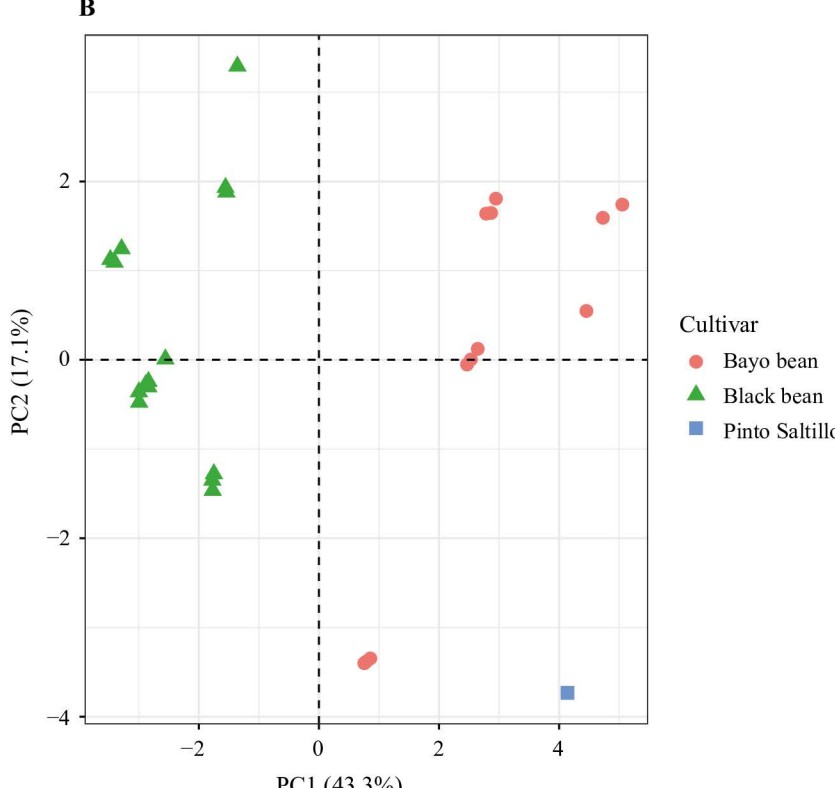

**Fig 3. Differences among common bean cultivars.** (A) Common bean seeds appearance of distinct *P. vulgaris* cultivars from INIFAP-Zacatecas germplasm bank. (B) PCA analysis of the phenological and morphological characteristics of plants and beans of the common beans cultivars.

cultivars based on seed characteristics such as size, shape, and pigmentation (Fig 3A). These cultivars are commonly cultivated in the Zacatecas region and are collected directly from farmers by the INIFAP. The experiment was performed at Campo Experimental-INIFAP Zacatecas, Mexico. Variations in the phenological aspects of the plants and seeds indicated significant differences between the Black and Bayo cultivars, as shown by PCA analysis (Fig 3B; S2 Table).

The alpha diversity of the rhizosphere bacterial communities of the Black and Bayo bean cultivars was lower than that of the bulk soil community, indicating a similar response of the soil bacterial community at the flowering stage when the rhizosphere samples of each cultivar

were collected (S20 Fig). Furthermore, similar to the rhizosphere of the Pinto Saltillo cultivar, Pseudomonadota were enriched in the rhizospheres of the Black and Bayo bean cultivars. Overall, the taxonomic composition of the rhizosphere bacterial community at the genus level was similar between the cultivars and Pinto Saltillo, highlighting the abundance of the genus previously identified in the rhizosphere of Pinto Saltillo (Fig 2A). Only some bacterial genera (*Achromobacter*, *Caulobacter*, *Rhizobacter*, *Sinorhizobium*, and *Sorangium*) with abundances higher than 1% were present in at least one of the nine cultivars, although they were underrepresented in Pinto Saltillo (Fig 4A and S6 Table). Beta diversity based on Bray-Curtis dissimilarity revealed significant differences in the rhizosphere bacterial communities of the Pinto Saltillo cultivar compared with the other cultivars (Fig 4B, PERMANOVA P > 0.1). In contrast, no significant differences were observed between the Black and Bayo cultivars (S7 Table). Furthermore, the rhizosphere bacterial communities of the three cultivars were significantly different from the bulk soil bacterial community (Fig 4B, PERMANOVA P < 0.001). Although data dispersion could affect the PERMANOVA test, further beta dispersion tests showed moderate homogeneity of the data, suggesting that PERMANOVA reflects true differences due to the composition and abundance of bacterial communities (S14 Fig).

Nevertheless, the compositional structure of the metagenomic data could confound the differences between the origin of the samples (soil/rhizosphere) and the cultivars (Pinto Saltillo/ Black/Bayo). To address this issue, we obtained the log ratio of the abundance of taxa in the metagenomic data using the centered-log ratio (CLR) and performed beta diversity with the normalized Aitchison distance matrix (S10 Fig). The PERMANOVA tests indicated significant differences between bulk soil and rhizosphere communities and that Pinto Saltillo had a bacterial community significantly different from that of Bayo and Black cultivars, which were closely similar (S8 Table). Therefore, the results underscore the reliability of the conclusions despite the compositional challenges in the metagenomic data.

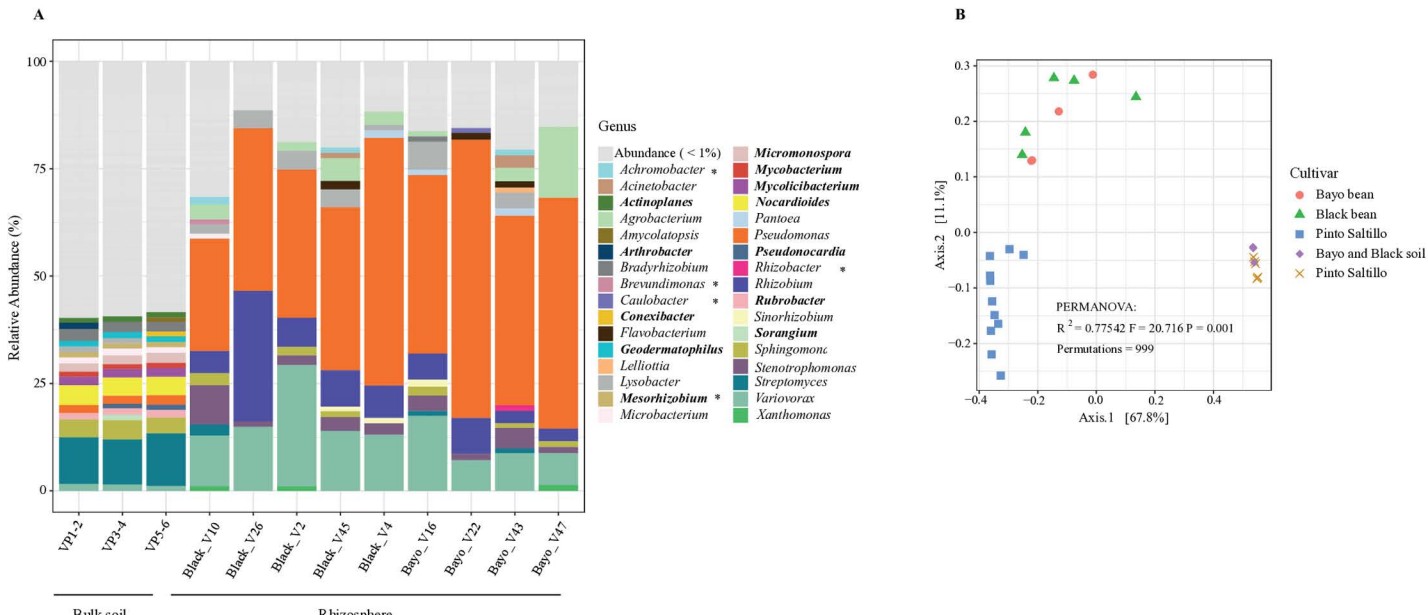

**Fig 4. Bacterial communities in different bean cultivars.** (A) Stacked bars with the Kraken2 taxonomic classification and abundance of sequence reads at the genus level from metagenomic samples of bulk soil and the rhizosphere of common bean cultivars. The right inset displays colors for genera with abundance > 1%. Genus names in bold are exclusive of bulk soil. Asterisks indicate genera that were underrepresented in Pinto Saltillo. (B) Bray-Curtis beta diversity analysis of the bacterial communities of the bulk soil and rhizosphere of common bean cultivars and Pinto Saltillo (PERMANOVA **p** < 0.001).

### Shared bacterial genera in the rhizosphere of common bean cultivars

To determine whether the rhizosphere effect on bacterial communities varied among the three common bean cultivars, we analyzed shifts in bacterial genera from the bulk soil to the rhizosphere using DESeq2 (Fig 5A-C). We focused on genera that showed more than two log2-fold enrichment in the rhizosphere based on genus-level rankings from Kraken2. In total, 88 genera were enriched in the rhizosphere of the three cultivars, including several genera related to PGPR, as previously discussed (Fig 5A–C). Although the rhizosphere of Pinto Saltillo had a greater number of unique genera, the rhizospheres of the Black and Bayo cultivars had fewer unique genera and were quite similar to each other, sharing 15 genera. These shared genera mainly belonged to the gamma and alpha classes of Pseudomonadota (Fig 5D and S9 Table).

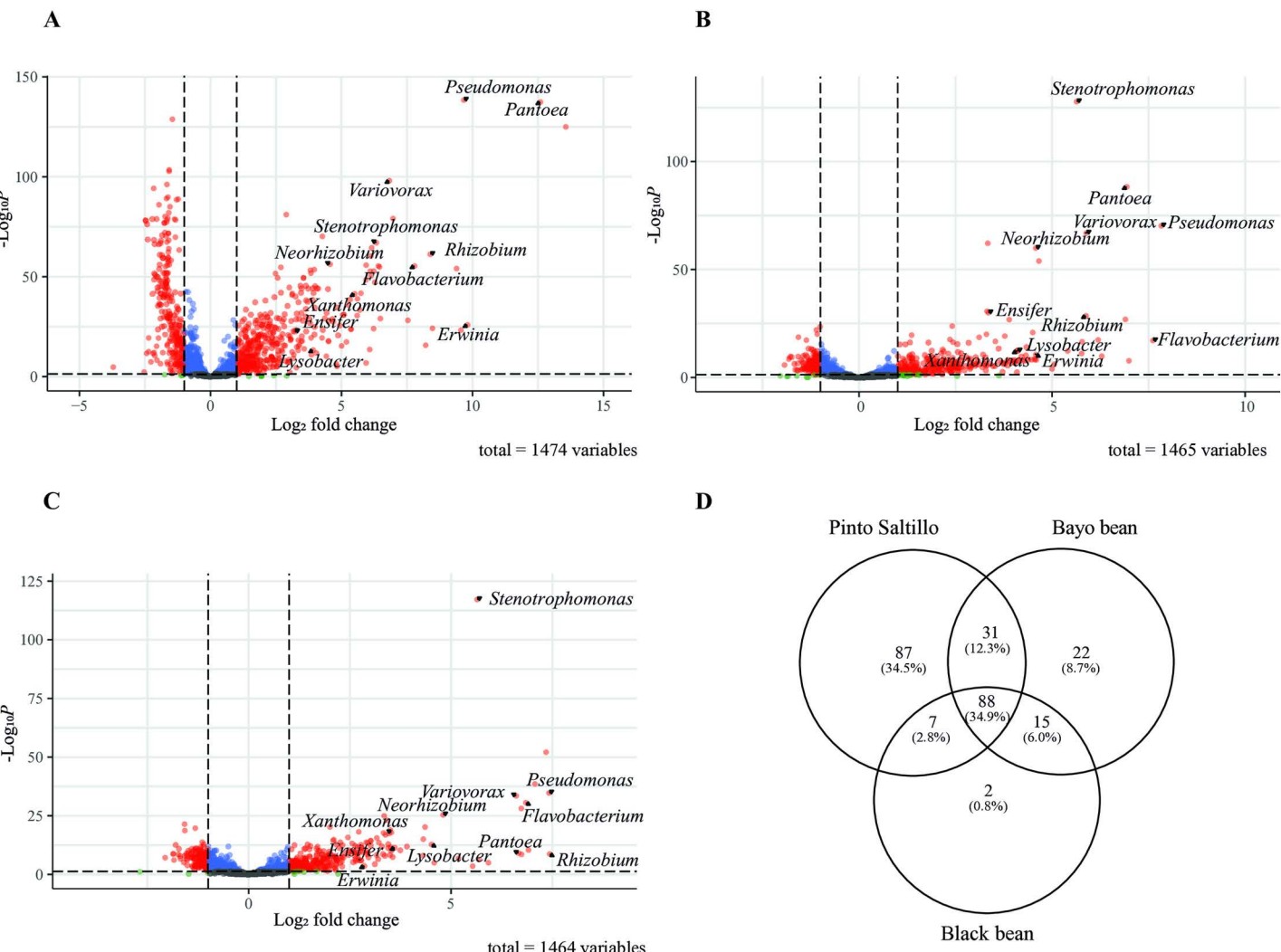

**Fig 5. Bacterial composition from the bulk soil to the rhizosphere.** Volcano plots of Pinto Saltillo (A), Bayo (B), and Black (C) common bean bulk soil to rhizosphere shifts calculated by DESeq. Differential abundance expressed as log2fold change (x-axis) and its statistical significance p-value (-log10, y-axis). OTUs significant enriched (p < 0.05) are represented by red dots. Parallel dashed lines delimited the area containing less significant changes (p > 0.05, log2-fold < 1); in blue are those non-significant OTUs that have a p-value < 0.05, but log2-fold < 1, and in gray are the non-significant OTUs that do not meet either of the two parameters (log2-fold < 1 and p-value > 0.05). (D) Venn diagram of selected OTUs enriched by more than a 2 log2-Fold Change.

These results emphasize the significance of the bulk soil community in the development of the rhizosphere community despite the variations among the different common bean cultivars.

## Discussion

In this study, we evaluated the bacterial communities in the bulk soil and rhizosphere of several varieties of common bean (*P. vulgaris*) cultivated in nearby agricultural regions. Our findings indicate that the diversity and structure of the rhizosphere bacterial community are predominantly influenced by the bulk soil bacterial community and, to a lesser extent, by the specific bean variety. This conclusion is supported by the experiments discussed below.

Initially, we performed an *in situ* sampling experiment in which bulk soil samples were collected from the site where the common bean seeds were sown (S1 Fig). In comparison with other studies, we carefully considered the definition of bulk soil as previous soil prior to planting [13,22]. For example, Pérez-Jaramillo [22] designed a greenhouse experiment using pots containing either native or agricultural soil to germinate wild and domesticated bean seeds. Pots without seeds served as the reference bulk soil. In contrast, Stopnisek and Shade [13] used non-adhered soil after vigorously shaking plant roots as the bulk soil. Although these designs were conceivably correct, our approach provided a more ecologically relevant characterization of the bulk soil bacterial community immediately before plant introduction.

Furthermore, an analysis of the bacterial diversity and composition across bulk soils of varying physicochemical compositions and with and without cultivation histories revealed no statistically significant differences in bacterial diversity (Figs 1, 2, and 4). This finding implies that the diversity of bacterial communities in bulk soil may be relatively stable under the field conditions in which they were collected.

Previous studies have noted a considerable effect of soil type on differences in bacterial rhizosphere composition in wild and modern common bean accessions in Colombia and Minnesota [13,22]. This dissimilarity may be attributed to the ample contrasts in physicochemical parameters observed between the Colombian and Minnesota soils sampled, including the pH (ranging from 4.7 to 8 for agricultural and native soils) and organic matter content [13,22]. Soil is often considered a critical contributor to microbial diversity in the rhizosphere of plants, and pH is a significant factor [46]. In *Rhizobium* legume symbiosis, pH plays a role in selecting the microsymbiont [47]. In acidic soils, *R. tropicii* dominates the nodulation of common beans, whereas in soils with neutral or near-neutral pH, *R. etli* and *R. phaseoli* are the most common [47–49]. In our experiment, slight variations in soil pH (by 0.4 units) and organic matter content might not have been significant enough to alter the bacterial communities in the bulk soils of plots A and N. Although it is well documented that the relative abundance and diversity of bacteria tend to increase as the pH shifts from 4 to 8, slight pH variations are not anticipated to exert an effect [46,50,51]. Long-term fertilization with nitrogen, phosphorus, and manure has been found to enhance the richness of the soil microbiota in alfalfa (*Medicago sativa*) but not in wheat (*Triticum*) [52]. Likewise, tillage and land use could influence the soil microbiome composition; however, in the agricultural plots studied here, any possible effects of these factors on the soil microbiome were indistinguishable.

In our study, we also demonstrated that the bacterial communities in the rhizosphere exhibited lower diversity compared to those in bulk soil (Fig 1). This observation aligns with previous findings in other plant species (7-9). In the Pinto Saltillo cultivar, cultivated in sites with and without agricultural antecedents, the bacterial rhizosphere displayed comparable diversity and composition (Fig 1). Furthermore, differential abundance analysis comparing bulk soil and rhizosphere communities of the Pinto Saltillo cultivar grown in sites with and without agricultural precedents revealed that 73% of the classified species according to

Kraken2 were similarly overrepresented in the rhizosphere (S16 Fig). These findings suggest that the bulk soil in plots with and without agriculture history may possess a common bacterial community that responds to plant presence while maintaining a group of specific taxa at each site.

Additionally, we investigated the bacterial communities in the rhizosphere of two distinct common bean cultivars, Black and Negro. The results revealed that, while the overall taxonomic composition of rhizosphere bacteria was similar across these cultivars, the Pinto Saltillo cultivar exhibited a significantly distinct bacterial community compared to the other cultivars, Black and Bayo, despite similar enrichment of Proteobacteria. Differences in the collection period and data processing for Pinto Saltillo, compared with the Black and Bayo beans collected and processed simultaneously, may account for the differences observed. However, examination of the overabundant genera shared among the three cultivars indicated 88 genera in common, with more than two-fold log fold change (Fig 5D).

Therefore, while soil and rhizosphere communities exhibited significant differences in diversity, rhizosphere composition was substantially influenced by the pre-existing soil bacterial community, whose diversity remained consistently similar under the studied field conditions. Nevertheless, the identification of a shared bacterial group within the community across cultivars and specific taxa responding uniquely to each cultivar suggests a balance between responses to soil and plant cultivars. Potential confounding effects attributable to differences in sampling periods and data processing may explain some of the observed differences in rhizosphere communities, particularly for the Pinto Saltillo cultivar. Despite these limitations, the shared patterns of enrichment among cultivars underscore the robustness of the conclusions.

Analysis of the rhizosphere bacterial community across various common bean cultivars revealed a significant enrichment of bacterial genera belonging to the phylum Pseudomonadota, which was consistently observed in all cultivars. Several genera associated with plant growth-promoting rhizobacteria (PGPR), including *Pseudomonas*, *Pantoea*, *Flavobacterium*, and *Rhizobium*, were identified. Notably, the common bean cultivar Pinto Saltillo influenced the proliferation of a group of similar bacterial species, irrespective of cultivation history or soil physical and chemical properties.

The dominance of Pseudomonadota in the rhizospheres of several plants has been extensively documented; however, it often shares a niche with Actinomycetota. In the Pinto Saltillo bean and the other nine cultivars studied here, the abundance of Pseudomonadota species was exceptional, representing more than 90% of the total and displaced species of the other phyla (S17 Fig). Studies based on high throughput 16S rRNA gene sequencing have shown that additional bacterial species of Acidobacteriota, Bacteroidota, and Verrucomicrobiota are regularly found in the rhizosphere of diverse bean cultivars [13,22,53]. In the rhizospheres of either Black or Bayo bean cultivars, a group of genera related to the Promoter-Growth Plant Rhizobacteria (PGPR), such as *Pseudomonas*, *Rhizobium*, *Pantoea*, and *Variovorax*, were the most abundant. Other genera, such as *Enterobacter*, *Lellotia*, and *Stenotrophomonas*, which are also related to PGPR organisms, were less abundant but were found in one or more rhizosphere metagenomic samples. Rocha et al. [54] recently reported a plethora of cultivated PGPR in the rhizosphere of *P. vulgaris L.*, the variety "Patareco" grown in Brazil, in agreement with the genus registered in this study.

The PGPR group has been intensively studied for decades because of its beneficial effects on plant growth and development, thereby offering opportunities for sustainable agriculture and biotechnological applications [55]. It is well known that *Pseudomonas* strains suppress pathogens in common beans [20] and nitrogen fixation by *Rhizobium* in symbiotic nodules [56]. Although we expected that *Rhizobium* species would be abundant in the rhizosphere of

common beans, the *Pseudomonas* genus consistently surpassed any other genus. Moreover, *Pseudomonas* was even more diverse, as 237 species were assigned to the genus, whereas *Rhizobium* had 45 species, according to Kraken2 (S19 Fig). Thus, *Pseudomonas* may have ample functional diversity in the rhizosphere of common bean at this agricultural site, such as growth promotion and protection against pathogens such as *Fusarium* and other fungi. Although the precise biological significance of the *Pseudomonas* abundance is unknown, this observation requires further experimental treatment. Other PGPRs, such as *Enterobacter*, *Pantoea*, and *Variovorax*, may produce phytohormones and antibiotics and contribute to phosphorus solubilization and root nutrient assimilation; however, these activities must be tested *in vitro* [55].

Pinto Saltillo is a cultivar developed using traditional agronomic breeding techniques and was selected for drought tolerance and resistance to rust, anthracnose, and common blight diseases. Pinto Saltillo is widely consumed in North-West Mexico, and the grain yield production per hectare is estimated at 1.5 tons/ha five years after its release [33]. It is less apparent what are the effects of plant hybridization, genetic manipulation techniques, and historical domestication practices on the rhizosphere microbiome diversity and function of common bean cultivars. This is a poorly studied area but comparing the rhizosphere microbiomes of *Fusarium*-resistant and *Fusarium*-susceptible common bean cultivars has revealed distinct rhizosphere microbiomes [20,29]. Additionally, rhizosphere microbiome composition depends on the root anatomy of the rhizosphere microbiome, which is probably modeled by domestication practices [19]. As pointed out by Fierer, "There is no "typical microbiome," instead, there is an ample variation in the relative abundance of taxa depending on the soil and plant genotype [46]. Therefore, the root microbiome of common beans is highly influenced by the agronomic features of the cultivar but is often neglected in breeding studies.

Our study contributes significant insights into the soil and rhizosphere communities of common bean (*P. vulgaris*) cultivars. Overall, these results offer an in-depth understanding of the rhizosphere in a limited selection of domesticated common bean cultivars and agronomic soils. However, these findings can be extended to encompass a broader range of common bean varieties and soil types and inform future field interventions in the form of biological agents for sustainable agriculture.

## Supporting Information

**S1 Supporting methods. Additional methods description.**
(PDF)

**S1 Fig. Location of the sampling site.** (A) sampling area within the INIFAP Experimental Field, in the state of Zacatecas, México (see geographic coordinates in S1 Table). Plots with agricultural and non-agricultural soil are indicated in the scheme. Nine additional bean cultivars were collected 1.9 km away from the sampling area geographic coordinates in S1 Table. (B) Experimental design of the sampling grid and location of collection sites. Soil and rhizosphere samples were taken at each intersection (yellow dots). Rhizosphere samples consisted of a rhizosphere pool of the soil root of 9 plants. The physicochemical analysis was performed with samples obtained at sites marked by blue dots. Physicochemical measures and soil classification was performed at INIFAP-Zacatecas.
(PDF)

**S2 Fig. Physicochemical characteristics of agriculture (A) and non-agriculture (N) soil.** Principal Component Analysis (PCA) of the soil chemical (pH; Nitrogen, N; Sodium, Na; Magnesium, Mg; Potassium, K; and Phosphorus, P) and physical (Organic Material, OM;

Field Capacity, FC; Saturation Point, SP; Permanent Wilting Point, PWP; Electric Conductivity, EC) properties. PERMANOVA statistic with Euclidian distance is shown below the plot.
(PDF)

**S3 Fig. Sequence reads classified by Kraken2 into the domains Bacteria, Archaea, and Eucaryota.** Sequence reads assigned to virus are also included. Kraken2 was used with the default parameters (confidence 0.0, minimum-hit-groups 2).
(PDF)

**S4 Fig. Classified sequence reads in rhizosphere and bulk soil metagenomes at different confidence scores (inset).** Metagenomic sequences came from bulk soil and rhizosphere of agriculture (A) and non-agriculture (N) soils of Pinto Saltillo common bean.
(PDF)

**S5 Fig. Bacteria taxa at species level classified utilizing Kraken 2 at different confidence scores (inset).** Metagenomic sequences came from bulk soil and rhizosphere of agriculture (A) and non-agriculture (N) soils of common bean Pinto Saltillo.
(PDF)

**S6 Fig. Shannon diversity index of bacteria communities in Bulk Soil (BS) and Rhizosphere of common bean Pinto Saltillo (R).** Shannon index was estimated with the taxonomic classification of Kraken 2 at different confidence scores (inset). T-test statistics between samples is shown.
(PDF)

**S7 Fig. Taxonomic classification with different databases** . (A) Comparison of the number of taxonomic units at genus level determined with Kraken2 using the ribosomal data bases RDP, GreenGenes, and Plus-PFP. (B) Number and percent of shared genera between Kraken2 classifications with the PlusPFP (KRAKEN2) and ribosomal data bases (GreenGenes, GREENG; and RDP, RDP_DTB).
(PDF)

**S8 Fig. U-Plots of the distribution of bacterial genus classified with Kraken2 using the data bases.** A. PlusPFP (n = 1464) B. RDP (n = 1484.) Most of the classified genus in A (95%) were shared in the metagenomes.
(PDF)

**S9 Fig. Rarefaction curves of bulk soil and rhizosphere metagenomes of distinct cultivated plants and common bean (this study).** Sample size is the number of metagenomic sequence reads in the rarefaction statistics. Species richness is the number of species classified by Kraken2. Green rectangle corresponded to the rarefaction of metagenomic sequences of bulk soil and rhizosphere of common bean in this study. A. All metagenomics samples without adjustment to the same sample size. B. All metagenomic sequences with the samples normalized to 20 million reads. Metagenomes *Arabidopsis*, bean-Mendes, cucumber, maize, tomato, and wheat raw metagenomic sequence were downloaded from GenBank (NCBI) and processed with the same procedures described in this work (see methods).
(PDF)

**S10 Fig. Beta diversity of bacterial communities of common bean cultivars (Pinto Saltillo, Black, and Bayo), from soil and rhizosphere.** Abundance matrix was calculated with the CLR approach and PCoA with the Aitchinson distances. Statistical analysis with PERMANOVA is shown in S8 Table.
(PDF)

**S11 Fig. Shared genera between Kraken and BRACKEN and beta diversity according to BRACKEN.** A. Venn diagram of shared genera classified by Kraken2 and BRACKEN (using the PlusPFP), and Kraken2 using the RPD. B. Beta diversity of bacterial communities of the common bean cultivar Pinto Saltillo, from bulk soil and rhizosphere. Abundance matrix was calculated with BRACKEN. PCoA shows Bray Curtis distances between samples.
(PDF)

**S12 Fig. Beta dispersion analysis of bulk soil and rhizosphere samples of Pinto Saltillo cultivar.** With the outlier sample AH (A, top left) and without the outliers (B). Beta dispersion distance to the centroid is shown in plots C and D. Statistical Levene´s test is shown in the table E.
(PDF)

**S13 Fig. Beta dispersion test of the metagenomic samples from bulk soil and rhizosphere of Bayo, Black, and Pinto Saltillo cultivars.** Levene´s test of homogeneity is shown below the figure.
(PDF)

**S14 Fig. ANCOM differential abundance.** Plots of the rhizosphere bacterial community cultivars A, Black bean. B, Bayo bean, and C, Pinto Saltillo. They were compared with the respective bulk soil bacterial community.
(PDF)

**S15 Fig. Common taxa at genus level differentially abundant (two log-fold) in the bacterial rhizosphere community, evaluated with DESeq2 and ANCOMBC** . A, Pinto Saltillo, B, Black bean, C. Bayo bean.
(PDF)

**S16 Fig. Differential abundance of the bacterial communities in the rhizosphere.** Cultivar Pinto Saltillo grown in Non-agricultural soil (A) and agriculture soil (B). Venn diagram shows the common taxa at species level, determined with Kraken2 (C).
(PDF)

**S17 Fig. Relative abundance of bacterial phyla presents in bulk soil and rhizosphere.** The stacked bars display 37 bacterial phyla. In bulk soil, the most prevalent phyla are Actinobacteriota (orange) and Pseudomonadota (green). Conversely, in the rhizosphere, Pseudomonadota comprise over 90% of the community.
(PDF)

**S18 Fig. Relative abundance of bacterial class in bulk soil and rhizosphere.** The stacked bars indicate the abundance of bacterial classes in bulk soil. Actinobacteria (red), Alpha-Proteobacteria (blue), and Beta-Proteobacteria (orange). Rhizosphere is represented by Gamma-Proteobacteria (light blue) and Alpha-Proteobacteria (blue), and Beta-Proteobacteria (orange). Bacterial classes with less than 1% abundance are shown in gray.
(PDF)

**S19 Fig Number of species classified using Kraken2, in selected enriched genera in the rhizosphere of Pinto Saltillo cultivar.** (A) *Rhizobium*, (B) *Pseudomonas*, and (C) *Variovorax*.
(PDF)

**S20 Fig. Alpha diversity of Black and Bayo bean cultivars.** Diversity indices calculated from the grouping of bulk soil and rhizosphere metagenomic samples of Black bean (n = 5)

and Bayo bean (n = 4). Wilcoxon test is equal to 0.05 for the differences between bulk soil and rhizosphere communities.
(PDF)

**S1 Table. Characteristics of the samples used for metagenome sequencing and GenBank accession data of bulk soil and rhizosphere of Pinto Saltillo, Black and Bayo common bean cultivars.**
(XLSX)

**S2 Table. Phenological and morphological characteristics of the Black, Bayo, and Pinto Saltillo common bean cultivars.**
(XLSX)

**S3 Table. Physicochemical properties of agricultural and non-agricultural soils.**
(XLSX)

**S4 Table. Statistics of metagenomic sequences analysis, and distribution of reads in domains determined by Kraken2.**
(XLSX)

**S5 Table. PERMANOVA statistics for beta diversity of bulk soil and rhizosphere bacterial communities in agriculture (A) and non-agriculture (N) plots, sown with Pinto Saltillo Cultivar.** PCoA was performed with Bray Curtis distances.
(XLSX)

**S6 Table. Differential abundance (DESeq2) of taxa at genus level from bulk soil and rhizosphere for Pinto Saltillo cultivar.**
(XLSX)

**S7 Table. PERMANOVA statistics for beta diversity of bulk soil and rhizosphere bacterial communities.** In agriculture (A) and non-agriculture (N) plots, sown with Pinto Saltillo. PCoA performed with Bray Curtis distances.
(XLSX)

**S8 Table. PERMANOVA statistics for beta diversity of bulk soil and rhizosphere bacterial communities.** In agriculture (A) and non-agriculture (N) plots, sown with Pinto Saltillo Black, and Bayo bean cultivars cultivars. Abundance tables were calculated with CLR and beta diversity with the Aitchinson distance.
(XLSX)

**S9 Table. Shared taxa at genus level in the Venn diagram of Fig 5.** Only genera enriched at LogFoldChange (LFC) > 2 in Pinto Saltillo, Black, and Bayo cultivars are shown.
(XLSX)

## Acknowledgements

Thanks to José Espíritu, Víctor del Moral, and Alfredo Hernández of the Unidad de Administración de Tecnologías de la Información (UATI), and Gabriela Guerrero and Luis Lozano of the Unidad de Apoyo Bioinformática (UAB), for bioinformatic assistance. Thanks to the undergraduate students José Luis Letechepia, Alejandro Espinoza, Juan Pablo López, and Jacob Romero for supporting soil collection, planting, and collecting bean plants at the INIFAP Experimental Field. Thank are also extended to Ivana Daysi Blancas Nava for help in laboratory materials. We are grateful to Dr. Juan Manuel Pichardo González, Curator Researcher

of the Centro Nacional de Recursos Genéticos, for the access to the seed germplasm used in this work, and Dr. Christian Sohlenkamp and the Centro of Ciencias Genómicas (CCG-UNAM) for academic and administrative support.

## Author contributions

**Conceptualization:** Víctor González.

**Data curation:** Griselda López Romo, Rosa Isela Santamaría.

**Formal analysis:** Víctor González, Jannick Van Cauwenberghe.

**Funding acquisition:** Víctor González.

**Investigation:** Griselda López Romo, Rosa Isela Santamaría, Víctor González.

**Methodology:** Griselda López Romo, Rosa Isela Santamaría, Patricia Bustos, Jannick Van Cauwenberghe.

**Resources:** Francisco Echavarría, Jannick Van Cauwenberghe.

**Software:** Patricia Bustos, Luis Roberto Reveles Torres.

**Supervision:** Rosa Isela Santamaría, Víctor González.

**Validation:** Víctor González, Jannick Van Cauwenberghe.

**Writing – original draft:** Griselda López Romo, Víctor González.

**Writing – review & editing:** Víctor González.

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
