## [Decision Letter · Decision Letter 0]

8 Oct 2024

PONE-D-24-23264The rhizosphere of domestic Phaseolus vulgaris L. cultivars hosts a similar bacterial community in local agricultural lands.PLOS ONE

Dear Dr. Lopez Romo,

Thank you for submitting your manuscript to PLOS ONE. After careful consideration, we feel that it has merit but does not fully meet PLOS ONE’s publication criteria as it currently stands. Therefore, we invite you to submit a revised version of the manuscript that addresses the points raised during the review process.

We look forward to receiving your revised manuscript.

Kind regards,

Kandasamy Ulaganathan

Academic Editor

PLOS ONE

Journal Requirements:

“PAPIIT-UNAM IN215908 for VG supported this study. GLR is a doctoral student from Programa de Doctorado en Ciencias Biomédicas, Universidad Nacional Autónoma de México (UNAM), and received a fellowship (No. CVU 746216) obtained from CONAHCYT. “

3. We note that [S1 Figure] in your submission contain [map/satellite] images which may be copyrighted. All PLOS content is published under the Creative Commons Attribution License (CC BY 4.0), which means that the manuscript, images, and Supporting Information files will be freely available online, and any third party is permitted to access, download, copy, distribute, and use these materials in any way, even commercially, with proper attribution. For these reasons, we cannot publish previously copyrighted maps or satellite images created using proprietary data, such as Google software (Google Maps, Street View, and Earth). For more information, see our copyright guidelines: http://journals.plos.org/plosone/s/licenses-and-copyright.

a. You may seek permission from the original copyright holder of S1 Figure to publish the content specifically under the CC BY 4.0 license. 

Reviewers' comments:

Reviewer's Responses to Questions

**Comments to the Author**

1. Is the manuscript technically sound, and do the data support the conclusions?

Reviewer #1: Partly

Reviewer #2: Yes

2. Has the statistical analysis been performed appropriately and rigorously? 

Reviewer #1: Yes

Reviewer #2: No

3. Have the authors made all data underlying the findings in their manuscript fully available?

Reviewer #1: Yes

Reviewer #2: Yes

4. Is the manuscript presented in an intelligible fashion and written in standard English?

Reviewer #1: Yes

Reviewer #2: Yes

5. Review Comments to the Author

Reviewer #1: I thoroughly enjoyed reading this manuscript. It was well written, justified, and organized and I commend the authors for providing a manuscript in such complete fashion.

However, before I can recommend any detailed changes the authors must provide some crucial information to the validity of the study related to their bioinformatics pipeline and subsequent diversity analyses.

First, for clarification the DNA was extracted from the rhizosphere soil, shotgun sequenced, and then run through a metagenomics pipeline that included using Kraken2 to taxonomically identify the reads. These taxonomic assignments were then used to conduct a diversity analysis of the samples. Is this correct?

These next questions will be made under the assumption the above pipeline is correct.

My major questions revolve around the validity of these taxonomic assignments based on WGS/shotgun reads.

1. How accurate/robust is the database from which these genes are taxonomically identified? The reason most people use 16S rRNA as a marker is that it is consistently present across all bacteria and is well annotated for species. Taxonomic ID's made from other functional genes (such as with shotgun metagenomic sequencing), are often much less resolved and thus your results are biased towards organisms with the most representation in the database. Please provide reasoning/justification for this process and evidence of reasonable non-biases for the datbase used.

2. Even if this method is valid and supported for taxonomic analysis, you provide no information about organisms other than bacteria, yet your taxonomic assignments were made across bacteria, fungi, protozoa, and plants. Did you remove those non-bacterial sequences prior to analysis, or were they left in? If left in, why do you only talk about the samples as if they contain bacteria? Please confirm or expand upon the methods used to generate "bacteria only" datasets from WGS.

3. A final question needed answering is the validity of diversity analysis made upon a gene profile that likely contains more than one representative of each individual organism. If organism A is highly abundant (and/or has a large genome), and you count all instances of a reads mapping to organism A, you're going to have a much larger sample of organisms A genes compared to a less abundant, or smaller genome-size organism, thus inflating the prevalence of organisms-A. Please provide justification or more detailed methods regarding how the data was handled in post-processing and bioinformatics.

Reviewer #2: "The rhizosphere of domestic Phaseolus vulgaris L. cultivars hosts a similar bacterial community in local agricultural lands"

The authors have made a valuable contribution to a topic of great significance of local and global relevance, presenting intriguing results in a well-organized manner. However, considering that it is a metagenomic study based only on read-based taxonomic assignment using Kraken2, it appears somewhat simplistic and represents a missed opportunity to leverage the extensive information generated for further analysis. For instance, examining other groups of organisms, such as eukaryotes and viruses, would be valuable. Additionally, enhancing taxonomic assignment by utilizing contigs instead of a read-based approach or conducting functional potential predictions after assembly could provide deeper insights, potentially leading to the reconstruction of genomes from metagenomes. It is important to note that read-based assignments can sometimes be overestimated. Considering the work as it stands and in order to improve quality and validate the results obtained, it is advisable to revisit the methodology to incorporate additional tools and methods that consider the nature of HTS data. Additionally, including the software and packages versions used would enhance transparency. Several suggestions have been proposed to further enhance the overall quality of the manuscript.

Abstract:

• The abstract is well-presented and organized.

Introduction:

• L59 and L62: Fabaceae are a family. Only genera and species must be in italic. Also, this line should be rewritten considering this, because Fabaceae are not species.

• The introduction should include information on Cultivation History and its impact on bacterial communities.

Material and methods

• L136: More details about physical and chemical characterization are needed in the methods section.

• L145: Is this analysis a PCA or a PCoA? Additionally, to assess differences between sites, a statistical test, such as a t-test or Wilcoxon test (depending on the normality of the data), could be employed.

• L159: Sequences “were” instead of “was”.

• L162: If the PlusPFP database was utilized, it would be important to clarify why other groups of organisms, such as viruses and eukaryotes, were not included in the analysis, particularly given that a metagenomic read-based taxonomic assignment was performed.

• L164: Considering that depth of sequencing could potentially be an issue; it is unclear whether a rarefaction or any type of normalization was conducted or if only rarefaction curves were generated. Providing a repository with scripts and analyses would greatly assist readers in understanding the procedures, methods and statistical analyses employed.

• Also, when Kraken2 is used, it is recommended to re-estimate the abundance with Bracken program (Lu et al., 2017).

- Lu, J., Breitwieser, F. P., Thielen, P., & Salzberg, S. L. (2017). Bracken: estimating species abundance in metagenomics data. PeerJ Computer Science, 3, e104.

• L167-L170: I strongly recommend taking into consideration the inherent characteristics of High-Throughput Sequencing (HTS) data, which can be both spurious and compositional due to the arbitrary total imposed by the instrument (Gloor et al., 2017). To address this issue, it is advisable to employ methods such as differential abundance tests, which can vary depending on the specifics of your research and the nature of your data (Nearing et al., 2022). Properly managing the data before hypothesis testing is essential to ensure the robustness of your results.

- Gloor, G. B., Macklaim, J. M., Pawlowsky-Glahn, V., & Egozcue, J. J. (2017). Microbiome Datasets Are Compositional: And This Is Not Optional. Frontiers in Microbiology, 8, 2224. https://www.frontiersin.org/article/10.3389/fmicb.2017.02224

- Nearing, J. T., Douglas, G. M., Hayes, M. G., MacDonald, J., Desai, D. K., Allward, N., Jones, C. M. A., Wright, R. J., Dhanani, A. S., Comeau, A. M., & Langille, M. G. I. (2022). Microbiome differential abundance methods produce different results across 38 datasets. Nature Communications, 13(1). https://doi.org/10.1038/s41467-022-28034-z

• Also, cite and detail packages of R used (packages instead of programs) with versions and properly citations.

• L171: It is advisable to use the term “statistical tests” rather than “statistics tests” to ensure clarity and precision

Results

• L185: Species and genera names always in italics.

• L202: Are these OTUs? According to the literature, OTUs are defined as clusters of sequencing reads that differ by less than a fixed dissimilarity threshold (Callahan et al., 2017). Furthermore, Kraken2 is not a clustering algorithm program but rather a sequence classifier. It utilizes k-mers and “de Bruijn” graphs to efficiently classify sequences by comparing them to a reference database of known sequences.

 Callahan, B. J., McMurdie, P. J., & Holmes, S. P. (2017). Exact sequence variants should replace operational taxonomic units in marker-gene data analysis. The ISME journal, 11(12), 2639-2643.

• What taxonomic unit or level does Kraken2 provide? It is important to standardize this terminology throughout the text for consistency.

• In Figure 1, the point that is distance to the others could be potentially an outlier? Consider the possibility and how to manage this type of data. Also, given the findings presented, it is recommended to conduct a betadisper analysis to validate the results of the perMANOVA. This additional analysis could provide insights into the dispersion of diversity within the groups.

• L209: The results of the PERMANOVA should be presented in detail, including not only the p-value but also the statistic, degrees of freedom, etc. This information could also be included in detail in a supplementary table.

• In figure 2 and 4, The colors are similar in some genera, making it difficult to distinguish between them.

• L226: All genus and species names must be italicized throughout the text. This error is consistent, as some are italicized while others are not.

• L240: DESeq2 results are in Figure 2.

• Results of the effect cultivation history in perMANOVA, alpha diversity and DESeq2 results are not presented. I suggest that these not significant values and results should be presented in order to express and discuss whether cultivation history has relevance or not in this study.

• L286: Figure S3 does not show data about Black and Bayo cultivars.

• In Figure 4B, it is evident that bulk soils are similar regardless of the cultivar type. However, this is not the case in the rhizosphere, where Pinto Saltillo samples show notable differences compared to the Black and Bayo samples. This raises concerns about potential confounding effects. For instance, was the impact of sampling time or seasonality accounted for, given that samples were taken in two different years and months? Additionally, were the sequences processed in two separate batches? (Sequencing and bioinformatic analyses) These factors could introduce confounding variables that might explain the apparent differences, especially since it was noted earlier that the abundance of genera was relatively similar across cultivars. This also applies to perMANOVA and DESeq2 analyses.

Discussion

• The overall structure of the discussion is well-organized

• L320: “Our findings indicate that the diversity and structure of the rhizosphere bacterial community are predominantly influenced by the bulk soil bacterial community, and to a lesser extent, by the specific bean variety.” This statement is unclear in terms of how the results support it. Additionally, it seems inconsistent with the statement in L341: “This study observed no significant differences in bacterial diversity across bulk soils of varying physicochemical compositions and cultivation histories.” Please clarify how the data support the conclusion that the rhizosphere is predominantly influenced by the bulk soil bacterial community, especially in light of the observation of no significant differences in bulk soil diversity.

• L396: The term “improved microbiome” is vague. Does it refer to greater richness or diversity, or is it about promoting the presence of specific types of bacteria? It would be helpful to define what is meant by "improved microbiome" in this context.

6. PLOS authors have the option to publish the peer review history of their article (what does this mean? ). If published, this will include your full peer review and any attached files.

**Do you want your identity to be public for this peer review?** For information about this choice, including consent withdrawal, please see our Privacy Policy .

Reviewer #1: No

Reviewer #2: No

---

## [Author Response · Author response to Decision Letter 0]

30 Dec 2024

Review Comments to the Author

Reviewer #1: I thoroughly enjoyed reading this manuscript. It was well written, justified, and organized and I commend the authors for providing a manuscript in such complete fashion.

However, before I can recommend any detailed changes the authors must provide some crucial information to the validity of the study related to their bioinformatics pipeline and subsequent diversity analyses.

First, for clarification the DNA was extracted from the rhizosphere soil, shotgun sequenced, and then run through a metagenomics pipeline that included using Kraken2 to taxonomically identify the reads. These taxonomic assignments were then used to conduct a diversity analysis of the samples. Is this correct?

R. Thank you for your comments. This is a summary of the experimental design. The taxonomic lower-level taxonomic classification of Kraken2 (species level) was used to model the rarefaction and diversity analysis (l. 185- 215; Supporting Methods)

These next questions will be made under the assumption the above pipeline is correct.

My major questions revolve around the validity of these taxonomic assignments based on WGS/shotgun reads.

1. How accurate/robust is the database from which these genes are taxonomically identified? The reason most people use 16S rRNA as a marker is that it is consistently present across all bacteria and is well annotated for species. Taxonomic ID's made from other functional genes (such as with shotgun metagenomic sequencing), are often much less resolved and thus your results are biased towards organisms with the most representation in the database. Please provide reasoning/justification for this process and evidence of reasonable non-biases for the datbase used.

R. In this study, we aimed to capture a broad range of genomic sequences to determine the diversity of the bacterial taxa. We utilized Kraken2 in our analysis with the PlusPFP database because of its robustness for broad-spectrum taxonomic classification of microbiomes (1-3). PlusPFP encompasses a wide range of microorganisms, expanding upon the Standard Database, which includes archaeal, bacterial, viral, and human sequences from NCBI RefSeq. Furthermore, PlusPGFP incorporated supplementary reference sequences for protozoa, fungi, and plants. We anticipated covering and discarding contaminant sequences from both plants and humans. Additionally, we expected sequence information for eukaryotes (fungi) and viruses; however, they were poorly represented in the metagenomes (see response 2 below) and were not utilized in our analysis. Conversely, the bacterial domain exhibited the highest number of sequences reads and diversity in metagenomes (S2 Table; S3 to S5 Figs.)

Several studies have addressed the performance of Kraken2 using different confidence parameters in simulated and mock databases of various sizes (2, 4). Kraken2 demonstrates efficacy in classifying sequence reads using a large database, similar in size to PlusPFP, at a default confidence score of zero (4). At this confidence score, a satisfactory recall of taxa remained, approximating the optimum of 0.69 (with a confidence score of 0.10). This indicates that these parameters accurately identified true-positive taxa. However, at a default score of zero, the overall performance of F1 is considered suboptimal (4).

Considering these parameters, we evaluated the effect of the default confidence score (zero) and three more stringent confidence score parameters (0.2, 0.4, and 0.6) on a sample of metagenomes derived from the bulk soil and rhizosphere. A consistent decrease in the number of classified sequence reads was observed as the confidence threshold increased from the default value of 0 to 0.6, accompanied by a corresponding increase in unclassified reads (S4 Fig.). In both the bulk soil and rhizosphere, the classified reads assigned to species diminished substantially, along with the diversity parameters (S5 and S6 Figs.). The Shannon index also exhibited low values at a strict confidence level threshold (S6 Fig). . Furthermore, the effect on classified reads was more pronounced in bulk soil than in the rhizosphere. This result led us to conclude that stringent confidence levels did not capture the full diversity of the sample, and significant taxa could be excluded because of their low representation, such as in bulk soil. Consequently, we opted to utilize the default Kraken2 parameters and subsequently examined the changes in abundance between the soil and rhizosphere to assess whether poorly represented taxa in the soil transitioned to abundance in the rhizosphere.

In the second instance, we evaluated the performance of Kraken2 using ribosomal RNA gene databases, specifically GreenGenes and RDP (5). In S7 Fig. shows comparisons of taxonomic assignments made using Plus-PFP, GreenGenes, and RPD databases. A total of 55% (808/1464) of the genera classified using Kraken2-PlusPFP were shared with the Kraken2-RDP database (S7 Fig.). Conversely, the number of classified genera in the GreenGenes database was less than that with RDP and shared approximately 38% (564/1464) with the classification with Kraken2-PlusPFP. Furthermore, the classifications with RDP were sparsely distributed among the metagenomes, in contrast to the homogeneity observed in the distribution of the Kraken 2-PlusPFP assignments across all 29 metagenomes (S7 Fig.).

Therefore, we concluded that the use of Kraken2 with the PlusPFP database provides a robust and diverse taxonomic classification for metagenomic sequences, especially in the bacterial domain. However, the application of stricter confidence parameters significantly reduces the taxonomic diversity detected, which may exclude important taxa with low representation, as observed in soil samples. Therefore, the default parameters of Kraken2 were preferred to avoid the loss of taxonomic diversity. However, the use of ribosomal gene databases (GreenGenes and RDP) with Kraken2 can complement the classification, although they have limitations compared to PlusPFP owing to differences in the coverage and distribution of taxa in the metagenomes. These discrepancies deserve further analysis, and detailed experiments should be performed with combined databases to exhaustively examine bacterial diversity in the soil and rhizosphere (3, 4).

This material and methods are available in the additional file Supporting Methods.

1. Wood, D.E., Lu, J. & Langmead, B. Improved metagenomic analysis with Kraken 2. Genome Biol 20, 257 (2019). https://doi.org/10.1186/s13059-019-1891-0

2. Lu J, Rincon N, Wood DE, Breitwieser FP, Pockrandt C, Langmead B, Salzberg SL, Steinegger M. Metagenome analysis using the Kraken software suite. Nat Protoc. 2022 Dec;17(12):2815-2839. doi: 10.1038/s41596-022-00738-y.

3. Edwin NR, Fitzpatrick AH, Brennan F, Abram F, O'Sullivan O. An in-depth evaluation of metagenomic classifiers for soil microbiomes. Environ Microbiome. 2024 Mar 28;19(1):19. doi: 10.1186/s40793-024-00561-w.

4. Wright RJ, Comeau AM, Langille MGI. From defaults to databases: parameter and database choice dramatically impact the performance of metagenomic taxonomic classification tools. Microb Genom. 2023 Mar;9(3):000949. doi: 10.1099/mgen.0.000949. PMID: 36867161; PMCID: PMC10132073

5. Lu J, Salzberg SL. Ultrafast and accurate 16S rRNA microbial community analysis using Kraken 2. Microbiome. 2020 Aug 28;8(1):124. doi: 10.1186/s40168-020-00900-2. PMID: 32859275; PMCID: PMC7455996.

2. Even if this method is valid and supported for taxonomic analysis, you provide no information about organisms other than bacteria, yet your taxonomic assignments were made across bacteria, fungi, protozoa, and plants. Did you remove those non-bacterial sequences prior to analysis, or were they left in? If left in, why do you only talk about the samples as if they contain bacteria? Please confirm or expand upon the methods used to generate "bacteria only" datasets from WGS.

R. We apologize for the inadequate description of this aspect of our research. It is pertinent to clarify (which has now been emphasized in the manuscript, l. 237-252; Supporting Methods) that our focus was on studying the diversity and composition of the bacterial community in bulk soils and in the rhizospheres of common bean cultivars. Consequently, we selected a DNA extraction kit capable of managing a wide range of bacteria but not specifically designed for eukaryotes, such as fungi. DNA samples were obtained using the Qiagen DNeasy PowerSoil Kit (Qiagen). Although the manufacturer stated that the kit is also suitable for fungi, in our experience, it is poorly represented (S4 Table).

In our experiments, a total of 12-14 Gb was sequenced for each metagenomic sample from the bulk soil and the rhizosphere. Most of the classified readings belonged to bacteria (97% and 99% for the bulk soil and rhizosphere, respectively), archaea (0.42, 0.01%), Eukarya (1.2, 0.2%), and viruses (0.02, 0.01%). Bacteria were the most abundant domains and the focus of this study (S4 Table; S3 Fig). Sequence reads other than bacteria, such as eukaryotes, archaea, and viruses, were excluded from further analysis using PhyloSeq.

3. A final question needed answering is the validity of diversity analysis made upon a gene profile that likely contains more than one representative of each individual organism. If organism A is highly abundant (and/or has a large genome), and you count all instances of a reads mapping to organism A, you're going to have a much larger sample of organisms A genes compared to a less abundant, or smaller genome-size organism, thus inflating the prevalence of organisms-A. Please provide justification or more detailed methods regarding how the data was handled in post-processing and bioinformatics.

R. It is acknowledged that several factors may potentially inflate the abundance count in Kraken2. The inflation of kmers in Kraken2 due to the genome length of the organisms likely introduces a bias in the abundance counts. This may also be influenced by the representation of an organism in the database. This issue was addressed by Lu et al. (2009) (6), through Bayesian Re-estimation of Abundance after Classification with Kraken2, or BRACKEN. Furthermore, BRACKEN estimated species despite their close relatedness. We concur with the reviewer that read counts can be assigned as the most represented. To address this, as suggested by Reviewer # 2, we re-analyzed Kraken2 data using BRACKEN. No differences were observed between the composition and abundance of taxa at the genus level between Kraken2 and BRACKEN (S11 Fig.). Additionally, we performed the beta diversity analysis using the abundance matrix of the bulk soil and rhizosphere of Pinto Saltillo (S11B Fig.) calculated with Bracken. This indicates that there were no differences with respect to the same analysis performed with the Kraken2 matrix (Fig. 1B).

6. Lu, J., Breitwieser, F. P., Thielen, P. & Salzberg, S. L. Bracken: estimating species abundance in metagenomics data. PeerJ Comput. Sci. 3, e104 (2017).

Reviewer #2: "The rhizosphere of domestic Phaseolus vulgaris L. cultivars hosts a similar bacterial community in local agricultural lands"

4. The authors have made a valuable contribution to a topic of great significance of local and global relevance, presenting intriguing results in a well-organized manner. However, considering that it is a metagenomic study based only on read-based taxonomic assignment using Kraken2, it appears somewhat simplistic and represents a missed opportunity to leverage the extensive information generated for further analysis. For instance, examining other groups of organisms, such as eukaryotes and viruses, would be valuable. Additionally, enhancing taxonomic assignment by utilizing contigs instead of a read-based approach or conducting functional potential predictions after assembly could provide deeper insights, potentially leading to the reconstruction of genomes from metagenomes. It is important to note that read-based assignments can sometimes be overestimated. Considering the work as it stands and in order to improve quality and validate the results obtained, it is advisable to revisit the methodology to incorporate additional tools and methods that consider the nature of HTS data. Additionally, including the software and packages versions used would enhance transparency. Several suggestions have been proposed to further enhance the overall quality of the manuscript.

R. Thank you for your insightful comment. We revised the manuscript in accordance with your concerns and suggestions. Indeed, we covered the extensive metagenomic information requested and performed compositional analysis of the HTS data according to the suggested methods. We have addressed all these points in the revised version of the manuscript and in the responses to reviewer #1. Versions of bioinformatic tools have been cited.

We conducted analyses of viruses and reconstructed MAGs from the contigs. These components were not included in the manuscript, as we focused exclusively on bacteria.

Another reason for excluding viruses and eukaryotes is the low abundance of these domains in the bulk soil and rhizosphere, as described in response to query 1 (reviewer 1; S3 Fig.; S4 Table). Regarding viruses, metagenomic data did not reveal the full extent of diversity. Rarefaction curves demonstrated that the viral readings were limited (Fig. 1 at the end of this document). A thorough examination of viral diversity warrants an experimental approach for the enrichment of viral readings. Santos-Medellín et al. published a comparison of viral metagenomic reads versus enrichment of the viral component in the rhizosphere of tomato (7). This does not imply that we disregarded the viral components. Our limited data indicate that viral diversity is correlated with the most abundant bacterial genera in samples. For instance, viruses associated with Pseudomonas, Rhizobium, and Flavobacterium, the most abundant genera classified by Kraken2, were the most abundant phages detected in the viral metagenomic viral fraction. However, these findings are insufficient to draw conclusions regarding the interplay between viruses and bacteria. Ongoing experiments focus on obtaining a more suitable representation of viruses using both viral cultivation and viral metagenomic enrichment approaches.

MAGs were reconstructed from contigs assembled from metagenomic readings; however, much of this work is still in progress. MAGs were consistent with the most abundant taxa determined by Kraken2 (Figure 2 at the end of this document). Here, we present some examples of MAGs coverage. We intended to describe the functional structure of the bulk soil and rhizosphere in another study, including details of the reconstruction of MAGs.

7. Santos-Medellin C, Zinke LA, Ter Horst AM, Gelardi DL, Parikh SJ, Emerson JB. Viromes outperform total metagenomes in revealing the spatiotemporal patterns of agricultural soil viral communities. ISME J. 2021 Jul;15(7):1956-1970. doi: 10.1038/s41396-021-00897-y.

Abstract:

• The abstract is well-presented and organized.

R. Thanks.

Introduction:

• L59 and L62: Fabaceae are a family. Only genera and species must be in italic. Also, this line should be rewritten considering this, because Fabaceae are not species.

R. You are correct, and this phrase is incorrect. It was modified as follows: “Nitrogen fixation symbiosis between Rhizobium and diverse leguminous species of the Fabaceae family is an outstanding example of the close relationship between plant roots and bacteria’ (l. 65- 67).

• The introduction should include information on Cultivation History and its impact on bacterial communities.

R. A new paragraph introducing the effects of cultivation history and other agricultural factors is now in the manuscript version (l. 79- 93).

Material and methods

• L136: More details about physical and chemical characterization are needed in the methods section.

R. We have now included a reference to the methods according to the Mexican Official Standard (NOM-021-RECNAT-2000). This is in l. l60- 162: “The physicochemical properties of the soil were analyzed at the INIFAP Zacatecas Soil Laboratory following the protoco

---

## [Decision Letter · Decision Letter 1]

10 Jan 2025

PONE-D-24-23264R1The rhizosphere of Phaseolus vulgaris L. cultivars hosts a similar bacterial community in local agricultural lands.PLOS ONE

Dear Dr. Lopez Romo,

Thank you for submitting your manuscript to PLOS ONE. After careful consideration, we feel that it has merit but does not fully meet PLOS ONE’s publication criteria as it currently stands. Therefore, we invite you to submit a revised version of the manuscript that addresses the points raised during the review process.

We look forward to receiving your revised manuscript.

Kind regards,

Kandasamy Ulaganathan

Academic Editor

PLOS ONE

Journal Requirements:

Reviewers' comments:

Reviewer's Responses to Questions

**Comments to the Author**

1. If the authors have adequately addressed your comments raised in a previous round of review and you feel that this manuscript is now acceptable for publication, you may indicate that here to bypass the “Comments to the Author” section, enter your conflict of interest statement in the “Confidential to Editor” section, and submit your "Accept" recommendation.

Reviewer #2: All comments have been addressed

2. Is the manuscript technically sound, and do the data support the conclusions?

Reviewer #2: Yes

3. Has the statistical analysis been performed appropriately and rigorously? 

Reviewer #2: Yes

4. Have the authors made all data underlying the findings in their manuscript fully available?

Reviewer #2: Yes

5. Is the manuscript presented in an intelligible fashion and written in standard English?

Reviewer #2: Yes

6. Review Comments to the Author

Reviewer #2: Thank you for your efforts in considering all the suggestions. I believe the manuscript has significantly improved as a result.

I have two minor comments for further refinement:

1. When referring to R2 (R-squared), please ensure that the superscript formatting is applied.

2. Kindly review the updated guidelines for referring to bacterial phyla, as outlined in the following resource: NCBI Taxonomy Update on Prokaryotic Phyla.

7. PLOS authors have the option to publish the peer review history of their article (what does this mean? ). If published, this will include your full peer review and any attached files.

**Do you want your identity to be public for this peer review?** For information about this choice, including consent withdrawal, please see our Privacy Policy .

Reviewer #2: **Yes: ** Stephanie Hereira-Pacheco

---

## [Author Response · Author response to Decision Letter 1]

23 Jan 2025

Rebuttal letter.

Journal Requirements:

R. Reference list was revised and updated. Two references were added in response to the reviewer #2 comment on taxonomy (see below)

6. Review Comments to the Author

Reviewer #2: Thank you for your efforts in considering all the suggestions. I believe the manuscript has significantly improved as a result.

I have two minor comments for further refinement:

1. When referring to R2 (R-squared), please ensure that the superscript formatting is applied.

R. The notation of R-square was formatted according to the convention, as cited in the manuscript.

2. Kindly review the updated guidelines for referring to bacterial phyla, as outlined in the following resource: NCBI Taxonomy Update on Prokaryotic Phyla.

R. The taxonomy has been actualized according to your comment. The names of the phyla have been modified accordingly throughout the manuscript, figures, and supplementary figures and tables. Table S6 was also modified to include the new taxonomic ranks and its synonyms. Additional lines (l. 199-202) were added to the methodology to account for these changes, in agreement with the NCBI Taxonomy. Two additional references concerning these modifications have also been included.

References included:

39. Pallen MJ. The dynamic history of prokaryotic phyla: discovery, diversity and division. International Journal of Systematic and Evolutionary Microbiology. 2024;74(9):006508. doi:10.1099/ijsem.0.006508.

40. Göker M, Oren A. Valid publication of names of two domains and seven kingdoms of prokaryotes. International Journal of Systematic and Evolutionary Microbiology. 2024; 74(1):006242. doi: 10.1099/ijsem.0.006242

---

## [Decision Letter · Decision Letter 2]

29 Jan 2025

The rhizosphere of Phaseolus vulgaris L. cultivars hosts a similar bacterial community in local agricultural soils.

PONE-D-24-23264R2

Dear Dr. Lopez Romo,

We’re pleased to inform you that your manuscript has been judged scientifically suitable for publication and will be formally accepted for publication once it meets all outstanding technical requirements.

Kind regards,

Kandasamy Ulaganathan

Academic Editor

PLOS ONE

Additional Editor Comments (optional):

Reviewers' comments:

Reviewer's Responses to Questions

**Comments to the Author**

1. If the authors have adequately addressed your comments raised in a previous round of review and you feel that this manuscript is now acceptable for publication, you may indicate that here to bypass the “Comments to the Author” section, enter your conflict of interest statement in the “Confidential to Editor” section, and submit your "Accept" recommendation.

Reviewer #2: All comments have been addressed

2. Is the manuscript technically sound, and do the data support the conclusions?

Reviewer #2: Yes

3. Has the statistical analysis been performed appropriately and rigorously? 

Reviewer #2: Yes

4. Have the authors made all data underlying the findings in their manuscript fully available?

Reviewer #2: Yes

5. Is the manuscript presented in an intelligible fashion and written in standard English?

Reviewer #2: Yes

6. Review Comments to the Author

Reviewer #2: Just a few that do not need further revision:

L317 still says "Firmicutes" and "Bacteroidetes" instead of "Bacillota" and "Bacteroidota."

L391 still says "Proteobacteria."

7. PLOS authors have the option to publish the peer review history of their article (what does this mean? ). If published, this will include your full peer review and any attached files.

**Do you want your identity to be public for this peer review?** For information about this choice, including consent withdrawal, please see our Privacy Policy .

Reviewer #2: No

---

## [Editor Report · Acceptance letter]

PONE-D-24-23264R2

PLOS ONE

Dear Dr. Lopez Romo,

I'm pleased to inform you that your manuscript has been deemed suitable for publication in PLOS ONE. Congratulations! Your manuscript is now being handed over to our production team.

Kind regards,

on behalf of

Dr. Kandasamy Ulaganathan

Academic Editor

PLOS ONE